# Ventricular Arrhythmias and Myocardial Infarction: Electrophysiological and Neuroimmune Mechanisms

**DOI:** 10.3390/biomedicines13061290

**Published:** 2025-05-23

**Authors:** Meng Zheng, Zhen Zhou, Ke-Qiong Deng, Hanyu Zhang, Ziyue Zeng, Yongkang Zhang, Bo He, Huanhuan Cai, Zhibing Lu

**Affiliations:** 1Department of Cardiology, Zhongnan Hospital of Wuhan University, 169 Donghu Road, Wuhan 430071, China; zn005359@whu.edu.cn (M.Z.);; 2Institute of Myocardial Injury and Repair, Wuhan University, Wuhan 430071, China; 3Hubei Provincial Clinical Research Center for Cardiovascular Intervention, Wuhan 430071, China

**Keywords:** ventricular arrhythmias, myocardial infarction, sympathetic nervous system, immune system, crosstalk

## Abstract

Ventricular arrhythmias (VAs) after myocardial infarction (MI) are still one of the most important causes of cardiovascular death, though patients receive timely vascular recanalization and drug treatment. And it requires further exploring the mechanism and new therapeutics of VAs induced by MI. Here, we review the electrophysiological and neuroimmune mechanisms of VAs induced by MI. Immune cells are regulated by combining with neuroendocrine molecules released by the sympathetic nervous system (SNS), and, in turn, they modulate SNS both at the paraventricular nucleus of the hypothalamus and stellate ganglion by releasing cytokines or chemokines. In addition, ‘life essentials’ such as sleep, physiological health, and exercise can also influence cardiovascular health through neuroimmune mechanisms. Those factors and mechanisms provide us with new perspectives for understanding the occurrence and maintenance of VAs after MI. Exploring the crosstalk between electrophysiology and neuroimmunology will contribute to finding new therapeutics for VAs after MI.

## 1. Introduction

Myocardial infarction (MI) is the leading cause of death and disability in cardiovascular disease (CVD) [1]. Benefiting from revascularization and other therapies, the short-term mortality of acute myocardial infarction (AMI) has declined from over 30% to 5–8% [2]. However, the risk of cardiac arrest and sudden cardiac death (SCD) remains increased. Cardiac arrhythmia is a major cause of SCD [3]. Of the 420,319 admissions with AMI-Cardiogenic Shock, ventricular tachycardia (VT) (35%) and ventricular fibrillation (VF) (30%) affect AMI patients [4]. Patients who survived AMI within 48 h and experienced ventricular arrhythmias (VAs) had increased mortality [5]. Meanwhile, long-term follow-up studies reveal that survivors with a reduced left ventricular ejection fraction faced a 13.1% incidence of non-sustained VT, a 3.0% incidence of sustained VT, and a 2.7% incidence of VF within 1.9 ± 0.5 years [6]. While β-blockers remain first-line therapy and implantable cardioverter-defibrillators (ICDs) reduce SCD in chronic phases, current antiarrhythmic strategies show limited efficacy for acute/subacute phases due to drug toxicities and paradoxical proarrhythmic effects [7]. An in-depth exploration of the mechanisms of occurrence and maintenance of VAs after MI, as well as the search for new therapeutic targets, is a breakthrough in the existing treatment model for arrhythmias. Emerging evidence reveals complex pathophysiological interactions between electrophysiological mechanisms and the immune or autonomic nervous system play an important role in arrhythmias after MI; readers are referred to reviews by Grune J et al. [8] and Herring N. et al. [9]. Unlike existing syntheses, this review innovatively integrates recent breakthroughs in cardiac neuroimmunology and neuromodulation studies, emphasizing the crosstalk between electrophysiology and neuroimmunology, which is expected to provide both timely and transformative insights for VAs management paradigms.

## 2. Methods

The authors conducted a systematic search across three databases: PubMed, Web of Science, and Google Scholar. The search spanned from 1 January 2016, to 1 January 2025. The search terms included “myocardial infarction” (1), “myocardial ischemia” (1), “ventricular arrhythmias” (2), “ventricular tachycardia” (2), “ventricular fibrillation” (2), “inflammation” (3), “immune” (3), “autonomic nervous system” (4), “sympathetic nervous system” (4), “neuroimmune” (5), and “neuroinflammation” (5). The search strategy is “1 and 2 and 3” or “1 and 2 and 4” or “1 and 2 and 5”. In addition, important references from the search papers have been included. Articles related to the field of search terms were included in this review, except for articles not available in English.

## 3. Electrophysiological Mechanism of VAs Induced by MI

### 3.1. Two Causes and Three Elements

Generally speaking, abnormal impulse initiation and conduction are the two main pathological mechanisms of arrhythmia, and the interaction between substrate, trigger, and regulatory factors all contribute to the two pathological mechanisms. Myocardial ischemia provides substrates for abnormal initiation, trigger, and conduction. Meanwhile, regulatory factors, such as sympathetic nervous system (SNS) hyperexcitation, electrolyte disturbances, left ventricular dysfunction, and inflammation, can modulate those substrates.

### 3.2. Three Stages of MI

MI is divided into three stages, which help us guide the proper treatment of arrhythmia. Stage I (from the beginning to 30 min post-MI) consists of two sub-stages, Stage Ia, 2 to 10 min after MI, and Stage Ib, 15 to 30 min after MI. Stage II (from 1.5 to 72 h post-MI) is irreversible, and VAs occurring at the repaired phase of MI are Stage III (beyond 1 month after MI) [10].

#### 3.2.1. Stage I, AMI

Stage I was cut into two sub-stages based on the irreversible changes in ionic equilibrium after 10–15 min post-MI. The cell metabolism changes due to myocardial ischemia contribute to the unbalanced distribution of ions, including H^+^, Na^+^, K^+^, and Ca^2+^, both inside and outside of cells, finally changing the electrophysiological properties of the myocardium.

##### Electrophysiological Properties at the Cellular Level

Action potential (AP) of cardiomyocytes consists of five phases: Phase 0, the rapid depolarization phase; Phase 1, the early repolarization phase; Phase 2, the plateau phase; Phase 3, the rapid repolarization phase; and Phase 4, the rest phase (shown in Figure 1). The AP is elicited by excitable cardiomyocytes, usually sinoatrial node pacemaker cells at physiological conditions, due to their overdrive suppression mechanism. Those pacemaker cells raise the potential from −85 mV to −65 mV, which is enough to activate the fast sodium current (I_Na_), then the potential rapidly depolarizes to +30 mV (Phase 0). The velocity and amplitude of phase 0 are vital electrophysiological characteristics affecting the conduction activity. When the potential reaches −40 mV, the voltage-gated calcium channels (CaL) and delayed rectifier potassium channels (including slow-Ks and rapid type-Kr) open, preparing for the subsequent repolarization phase. The activation of potassium transient outward channels (K_to_) and the inactivation of sodium channels lead to the early repolarization phase (Phase 1). The activity of CaL reaches a peak at the beginning of the plateau phase; at the same time, the delayed rectifier potassium channels gradually increase their opening rate and peak at the end of the plateau phase; these two opposite ionic currents contribute to the plateau phase, where calcium influx is the hub current of electromechanical coupling of the myocardium. The potassium efflux forms Phase 3 depending on K_s_ and K_r_. Finally, with the help of type 1 potassium channels (K_1_), the potential is maintained at a stable polarized state: the resting membrane potential (RMP) of working ventricular and atrial myocytes (shown in Figure 1). Metabolites and H^+^ accumulated due to myocardial ischemia result in abnormal ion distribution, representing the elevation of extracellular potassium ion concentration and the increase of intracellular sodium and calcium ion concentration.

RMP, Velocity, and Amplitude of Depolarization

The depolarization of the RMP in the ischemic area has been demonstrated by direct measurement with intracellular glass microelectrodes. Firstly, the elevation of extracellular K^+^ contributes to the decline of RMP (shown in Figure 1). The accumulation of K^+^ is reversible within the early 15 min of ischemia, but as the occlusion continues, this change becomes irreversible. The Na^+^/K^+^-ATPase dysfunction, the development of intracellular acidosis, and potassium channels activated by the elevation of intracellular sodium concentration, all can contribute to the potassium efflux [11]. The extracellular high-K^+^ situation after MI leads to the suppression of I_Na_, which is voltage-gated. As RMP depolarizes, I_Na_ gradually undergoes inactivation. The capacity of Na^+^ fluxing into the intracellular determines the amplitude and upstroke velocity of AP (shown in Figure 1)**.** Finally, the cells become completely non-excitable while the RMP depolarizes to −60~−65 mV. Therefore, the extracellular high-K^+^ situation results in the restraint of impulse initiation. In addition, due to the limitation of oxygen supply, cardiac metabolism transforms from oxidative metabolism to anaerobic glycolytic. The degradation of membrane phospholipids can directly or indirectly interact with ion channels [12]. Therefore, the key mechanism of depolarization during the acute ischemic stage deserves to be further explored.

Action Potential Duration and Effective Refractory Period

The action potential duration (APD) is a vital electrophysiological property of cardiomyocytes, defined as the time from the onset of AP depolarization until the time of recovery to RMP. Theoretically, channels affecting the depolarization or repolarization of AP can change APD. During the first 2 min post-MI, APD temporarily lengthens, followed by continuous reduction during AMI. Correspondingly, the effective refractory period (ERP) of cardiomyocytes increases, followed by a reduction. However, ERP may not always synchronously change with the variation of APD during ischemia. Partially on account of the depolarization of RMP, ischemic cardiomyocytes may not recover promptly from non-excitable conditions, which leads to ERP prolonging, sometimes longer than APD. Phases 2 and 3 are the two vital phases affecting APD, in which potassium channels are responsible for the changes in APD during the ischemic phase. In the first 2 min, K_s_, K_r_, and K_to_ are all suppressed for intracellular acidification [13]. Besides, the metabolites from amphiphilic FA also contribute to reducing K^+^ efflux current [14]. As ischemia persists, cardiomyocytes exhaust ATP, and the elevation of APD activates K_ATP_ [11]. AA, originating from the FA metabolism, activates K_AA_ [11], and the increase of intracellular Na^+^ and Ca^2+^ activates K_Na+_ and K_Ca2+_ [11]. All of these increase K^+^ efflux, so the APD at the ischemic region decreases (shown in Figure 1). In addition, Na^+^ current also affects APD. The suppression of I_Na_ secondary to extracellular high-K^+^ concentration results in the reduction of amplitude and upstroke velocity of AP, which consequently contributes to the decrease in APD [11].

Trigger

Trigger activity is one of the two origins of abnormal automaticity, besides depolarization occurring at Phase 4. Generally, trigger activity manifests as afterdepolarizations (AD), which consist of two types: early afterdepolarization (EAD) and delayed afterdepolarizations (DAD), according to the time point of depolarization. EAD occurs at the repolarization phase, while DAD occurs after repolarization is completed or almost completed (shown in Figure 1). When the abnormal oscillation in membrane potential reaches the AP threshold, a new depolarization initiates, leading to a premature systole [15]. The occurrence of EAD depends on the prolongation of APD. Especially at the early stage of ischemia, the aforementioned ion currents contributing to APD prolongation predispose cardiomyocytes to be susceptible to initiating a premature systole. In addition, ROS prevents I_Na+_ from inactivation, and the increase of Na^+^ influx also aggravates the prolongation of APD [16]. Around 10–15 min after ischemia, sympathetic hyperactivity can promote Ca^2+^ influx by releasing catecholamines. Ca^2+^ overloading contributes to both EAD and DAD. Factors leading to increased Ca^2+^ influx to the cytoplasm and reduced Ca^2+^ efflux can elevate Ca^2+^ concentration in the cytoplasm. Physiologically, endoplasmic reticulum (SR) Ca^2+^ ATPase (SERCA) activity undertakes 70% of Ca^2+^ extrusion from the cell at diastole. While cells experience ischemia, ATP exhaustion, and ROS increase, all inhibit SERCA activity, leading to the decline of Ca^2+^ recycling into SR. Additionally, cardiac sympathetic hyperactivity promotes the phosphorylation of ryanodine receptors (RyR) by PKA and CaMKII, leading to Ca^2+^ leakage from the SR [17]. Moreover, the activation of NH current due to intracellular acidification increases Na^+^ influx, and then the influx of Ca^2+^ is reversely reinforced secondary to the elevation of Na^+^ concentration by NCX [18]. In addition, metabolites also affect the electrophysiological properties of cardiomyocytes, and researchers found that Purkinje fibers exposed to lysophosphatidylcholine exhibited EAD [19]. Stretch can also induce EAD [19]. Interestingly, EAD is suppressed in severe ischemic areas. At the same time, the blood in the ventricular cavity protects the Purkinje fibers adjacent to the ischemic region from ischemia. However, cardiomyocytes in the middle layer of the ventricle wall had relatively prolonged APD, making them the preferred trigger site of EAD [19].

**Figure 1 biomedicines-13-01290-f001:**
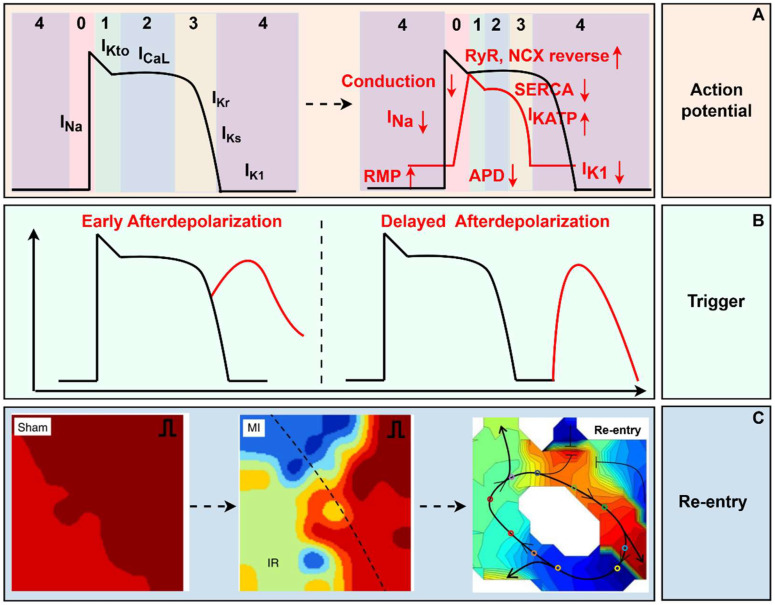
The electrophysiological mechanism of myocardial infarction. (**A**) RMP depolarized, and the amplitude and velocity of depolarization decreased, and APD shortened post-MI. (**B**) Early afterdepolarization occurs at the repolarization phase, while delayed afterdepolarization occurs after repolarization. (**C**) At the tissue level, reentrant ventricular arrhythmia originating from spatial heterogeneity and conduction block was recorded through optical mapping [20,21]. In the Sham and MI groups, red represents a short activation time, while blue represents a long activation time. In re - entry, red represents a long activation time, and blue represents a short activation time. RMP, rest membrane potential; RyR, ryanodine receptors; NCX, Na^+^/Ca^2+^-exchanger; SERCA, sarcoplasmic reticulum Ca^2+^-ATPase; APD, action potential duration; MI, myocardial infarction. (By Figdraw, https://www.figdraw.com/static/index.html#/, 11 March 2025).

Electrophysiological Properties at Tissue Level

The heart’s conduction is sequentially excited physiologically, guiding the heart to orderly contract and dilate and assuring full arterialization of venous blood pumped out of the left ventricle to supply organs. The aforementioned abnormal automaticity and trigger activity of injured cardiomyocytes must ascend to the tissue level to maintain sustained VAs. Regarding tissue level, the most remarkable features during the acute ischemic stage are inhomogeneity, injury currents, and conduction block.

Inhomogeneity

Inhomogeneity is reflected in spatial heterogeneity. Both ischemic and normal heart tissue possess inhomogeneity. Following the coronary artery obstruction, the heart is naturally divided into the infarcted region, the border zone, and the non-ischemic region. Tissue at the infarcted region experiences extracellular K^+^ rising, intracellular Na^+^ and Ca^2+^ elevation, RMP depolarization, APD changing, and ERP prolonging, while tissue at the border zone and non-ischemic region still receives adequate blood supply, so K^+^ moves from the infarcted region to the non-ischemic area. This heterogeneity may lead to differences in RMP, APD, and ERP. So, while the ERP at the central infarcted zone prolongs a lot due to the post-repolarization-refractoriness, the ERP at the border zone is shorter than that in a physiological situation. Transmural dispersion also exists. On the one hand, the subendocardial myocardium adjacent to the ventricular cavity is supplied directly from blood in the cavity; on the other hand, the middle myocardium has fewer I_ks_ and more I_Na_ and I_Na-Ca_, making it more susceptible to ischemic injury [19]. In addition, cardiac sympathetic nerves distribute inhomogeneity at different heart areas and ventricular layers [22]. Purkinje fibers also contribute to the inhomogeneity of the ischemic heart tissue because Purkinje fibers are more tolerant to ischemia than ventricular myocardium. In experiments, spatial electrophysiological heterogeneity can be quantified as the difference between the longest and shortest ERP, identified as the dispersion of ERP (DERP). Researchers found that DERP increased significantly in the whole heart by the S1–S2 stimulation method after MI. It is worth noting that ERP is diverse during the early ischemic stage, and many factors can modify it. Stimulating cardiac sympathetic nerves increased the dispersion of the global ventricular electrophysiological properties [23]. MI is often accompanied by heart failure, which manifests as an increase in left ventricular end-diastolic pressure. The elevation of ventricular pressure had been reported to increase arrhythmia inducibility for shortening ventricular APD and ERP, and those effects were spatially heterogeneous [24]. Therefore, the changes in ERP or DERP need careful discussion. Furthermore, electrophysiological heterogeneity is time-dependent, as the cardiac injury gets more severe and irreversible with time passing by. The spatial and time heterogeneity provides essential elements for injury currents, abnormal conduction, and re-entry, significantly increasing susceptibility to VAs after MI.

Injury Currents

Injury currents are generated by the differences in membrane potentials between the ischemic region and the normal myocardium. Injury currents were demonstrated in isolated dog hearts, showing that current flowed from depolarized tissue toward normal tissue [25]. Injury currents accompanied by trigger activity both contribute to inducing premature ventricular depolarizations after MI.

Conduction and Re-entry

Conduction activity of the heart depends on the excitability of the myocardium and cellular coupling. The former is influenced by the RMP, the amplitude and velocity of the AP, the APD, and the ERP of the injured myocardium. During the first 2 min post-MI, the RMP moderately depolarizes from −90 mV to −80 mV due to a mild increase in extracellular K^+^ concentration. Consequently, the conduction velocity increases. After the first 2 min, the conduction velocity decreased significantly in the ischemic region. Firstly, the severe depolarization of the RMP leads to inactivation of I_Na_, resulting in a decline in conduction velocity. Secondly, due to the injury myocardium taking more time to recover from inactivation, the ERP is prolonged much more, even though the APD gets shorter. Cx43 protein is the primary cellular coupling of the ventricle, and its activity, distribution, and expression influence the longitudinal conduction of the myocardium. The rapid cellular uncoupling occurs approximately 15 min post-MI, leading to an elevation of intercellular impedance [26]. Furthermore, changes associated with myocardial ischemia, such as Ca^2+^ overloading, intercellular acidification, elevation of FA and LPS metabolites, and overflow of catecholamines, all contribute to the dysfunction and reduction of Cx43 [27]. Those cellular and intercellular changes lead to decreased tissue conductance, providing a functional and structural basis for unidirectional block and re-entry trigger. The classic re-entry path consists of two pathways: one is slow conduction but has a shorter ERP, and the other is fast conduction but has a prolonged ERP. An impulse can form a whole re-entry circuit only at the right time. When the excitation impulse reaches a functional or structural obstacle, the impulse excites the former pathway until the latter pathway recovers from its ERP, and the impulse reverses transmission through the latter path. The big circuit can be shunted into a smaller ring if obstacles are dispersed, and then VT degenerates to VF. Clinically, patients with VT or VF occurring at the early ischemic stage are always pre-hospital, accounting for a large proportion of out-of-hospital cardiac arrest (OHCA). A retrospective study showed that 7.9% of patients with cardiogenic OHCA had initial fatal arrhythmia [28], suggesting the difficulty of preventing OHCA. Here, we realize one of the most essential electrophysiological mechanisms of VT and VF is re-entry. Thus, the reentrant initiation and sustaining mechanism is a breakthrough for searching for new strategies to decrease the occurrence of VT or VF at the early ischemic stage.

#### 3.2.2. Stage II, Subacute MI

Stage II is between 1.5 and 72 h after MI. The primary mechanisms of VAs at this stage are abnormal automaticity, which originates from the surviving Purkinje fibers, and re-entry activity [29]. Researchers contrasted ECG and hemodynamic characteristics between Stage I and Stage II, and found no significant differences in blood pressure, ECG intervals, and even the VA types were similar. However, the magnitudes of ECG’s R and Q waves at Stage II were lower than those at Stage I, suggesting that Stage II represents the evolution of Stage I and that similar arrhythmogenic mechanisms are involved in both stages [29]. VAs at phase II can manifest as ventricular premature, accelerated idioventricular rhythms, and VTs. In addition, cardiac SNS plays a crucial role in Stage II VAs. Ischemia-induced arrhythmia did not occur at Stage II in isolated rats’ hearts, suggesting the arrhythmogenesis of the autonomic nervous system (ANS) and/or circulating factors from blood in Stage II [30]. Notably, there are limited studies on Stage II VAs compared with Stage I. Clements-Jewery H had compared the effects of antiarrhythmic drugs (AADs) between Stage I and II, and most AADs were ineffective to VAs occurring at Stage II [29]; the author thought the arrhythmogenic effect of AADs could not be excluded. Stage II VAs need more attention and evidence to guide treatment.

#### 3.2.3. Stage III, Chronic MI

The chronic phase is also divided into two sub-phases, similar to the acute ischemic phase, where the former occurs within a month and the latter occurs a month later. Generally speaking, the originating site of VAs is the surviving myocardium, primarily located in the subepicardial layer. The re-entry activity is the primary electrophysiological mechanism, in which anisotropy was thought to be the possible cause of slow conduction, rather than depressed transmembrane potentials leading to slow conduction at the acute ischemic phase. At the cellular level, the maximum diastolic potential, amplitude, and V_max_ of phase 0 of the transmembrane potentials all decreased within the first week after ischemia. Meanwhile, the APD was also reduced. By the second week, all formerly mentioned properties, except the APD, have returned to normal. A month later, the characteristics of the transmembrane potentials are almost normal, except for the effective conduction velocity, which remains slow due to the anisotropy originating from myocardial fibrosis. Researchers used an optical mapping technique to synchronously record the re-entry circuit, confirming the slowing of conduction and the re-entry loop at the infarcted heart (shown in Figure 1) [31].

### 3.3. Conclusion of the Three Phases of MI

We discuss the three phases of MI because they correspond to the three clinical stages, including prehospital, in-hospital, and post-hospital stages. Although therapeutics for MI have been developed, the risk of cardiac arrest and SCD remains high in the first month, mainly originating from malignant VAs, and ICDs are the only way to save post-hospital deaths of MI. Although the prophylactic application of ICD reduced the mortality induced by arrhythmia, it did not improve the overall prognosis in high-risk MI patients [32]. So, the electrophysiological mechanism studies of MI are critical. In terms of electrophysiological mechanisms, re-entry activity and injury current contribute to VAs occurring at Stage Ia, while Stage Ib VAs arise from re-entry activity and abnormal automaticity. In comparison with Stage Ia, electrophysiological properties of subepicardial myocardium are inhomogeneous, and the reduction of amplitude and upstroke velocity of AP, and the shortening of APD are improved as ischemia propagates to 12–30 min. However, VT occurring at Stage Ib is easier to degenerate into VF. Conduction block caused by the dysfunction and decreased expression of intercellular gap link and the aggravation of APD shortening, the heterogeneity of apex-basal and transmural axis due to cardiac sympathetic overactivation, is the culprit of VAs. Stage II is the extension of Stage Ib, and their mechanisms are similar. Stage III VAs are relatively stable due to the repair of the heart. Factors regulating or reflecting the features of electrophysiological mechanisms may become the targets for intervention post-MI. Recently, factors regulating MI, like the immune system and ANS, have received much attention. The following parts will discuss these factors in MI.

## 4. Electroimmunological Mechanism of MI

Inflammation occurs as early as the first minutes after ischemia and persists throughout the entire process of MI, encompassing the phases of cardiac injury, proliferation, and repair. Proper inflammation promotes the removal of necrotic tissue during the early ischemic phase and facilitates cardiac repair during the later proliferation and repair phases. In the past, immunology was thought to be incidentally associated with arrhythmia. However, recent research revealed that cardiac immune cells could interact with cardiomyocytes directly or indirectly by changing the composition of the heart and altering the conduction of myocardial tissue, finally leading to arrhythmia. Many clinical investigations have demonstrated that inflammation is significantly correlated with VAs in MI [33]. In addition, clinical research has proved that patients with higher IL-6 concentrations were suitable candidates for receiving ICD treatment and were more likely to benefit from ICD implantation [34]. Preclinical studies demonstrated that immune cells can influence the excitability and conduction activity of the myocardium. Grune and his colleagues summarized those mechanisms as four electroimmunological pathways: leukocyte release of cytokines that act on cardiomyocytes; altered electrotonic gap junction communication between conducting cells and leukocytes; leukocyte-instigated, insulating fibrosis; and autoimmune channelopathies [8].

### 4.1. The Modulation of Inflammation on Cardiomyocytes

Cytokines like IL-6, tumor necrosis factor (TNF), IL-1, and IL-17 have been reported to prolong APD and impair intracellular Ca^2+^ handling [35]. The most well-investigated cytokine is TNF, which could inhibit potassium efflux, prolonging APD and QT interval. TNF reduced the expression or impaired the transport function of potassium channels, such as I_to_ [36], I_kr_ [37], and so on. The same effect was also obtained in IL-1 [38] and IL-6 [39]. The prolonged ventricular APD predisposes cardiomyocytes to be more susceptible to EAD. Another critical effect of inflammation on cardiomyocytes is intracellular Ca^2+^ handling. TNF [40], IL-1 [41], IL-6 [42], and IL-17 [43] had all been reported to increase intracellular Ca^2+^ concentration by increasing RyR2 expression and calmodulin-dependent protein kinase II oxidation, reducing the expression of SERCA and PLB phosphorylation, which resulted in DAD triggering. Through IL-1 receptor inhibitor treatment, intracellular Ca^2+^ handling was improved, and the magnitude of Ca^2+^ alternans decreased. Those changes were linked to the inhibition of SERCA expression.

### 4.2. The Modulation of Inflammation on Electrotonic Gap Junction

Regarding conduction, it is inspiring that Hulsmans and his colleagues have proven that macrophages directly connect with cardiomyocytes during in vivo and in vitro experiments. Then, they exclusively knocked out the Cx43 in macrophages, leading to a remarkable decrease in conduction in heart tissue. As a result, the heart was more susceptible to VAs than the control group [44]. Fibrosis is one of the most vital characteristics of a healed heart after MI. Fibrosis in the infarcted area protects the injured heart from rupture; however, overactive fibrosis, with fibrosis spreading beyond the infarcted region, leads to cardiac remodeling. Scientists have proven that microscopic fibrous foci, widely distributed at intercellular substances, are the histopathological features of arrhythmia [10].

### 4.3. The Modulation of Inflammation on Fibrosis

Fibrosis consists of two types: interstitial fibrosis and myofibroblast–cardiomyocyte coupling. The compact interstitial fibrosis at the infarcted area is non-excitable, leading to conduction block, subsequently provoking re-entrant arrhythmia. This re-entrant arrhythmia is stable, presenting as monomorphic VT at surface ECG, and can be terminated easily by low-energy force. Fibrosis at the border or in non-ischemic regions is more arrhythmogenic, and this type of fibrosis induces electric excitation out of control of source-sink mismatch and promotes ectopic triggers [45]. Myofibroblast–cardiomyocyte coupling leads to post-repolarization refractoriness, increases APD dispersion, and enhances re-entry potential [46]. Recently, researchers found that fibroblast depolarization through exclusively expressing the optogenetic cationic channel ChR2 in cardiac fibroblasts at scar areas elicited cardiac excitation and induced arrhythmia [47]. Collectively, fibrosis after MI has great arrhythmogenic potential by increasing the excitability of cardiomyocytes and aggravating conduction block. Indeed, factors like immune cells and their production are closely connected with fibrosis. In the early ischemic stage, monocytes are recruited to the infarcted region by CCR2/CCL2 signaling and then differentiate into macrophages to clear necrotic tissue [48]. Inflammation recedes gradually 3–5 days after MI, and the secretion of pro-inflammatory cytokines like IL-1 and TNF-α declines. At the same time, TGF-β and IL-10 are produced by macrophages, stimulating fibroblasts remodeling to myofibroblasts and then promoting collagen synthesis and deposition [49]. Researchers deleted cardiac fibroblasts’ IL-1 receptor and CCR^2+^ monocytes and macrophages’ IL-1β ligand and inhibited IL-1β signaling by a specific antibody, and results showed that myocardial fibrosis and cardiac function were well improved. These findings suggest that the crosstalk of macrophages and fibroblasts was mediated by IL-1β signaling, highlighting that targeted anti-inflammatory methods are potential therapeutics for cardiac fibrosis. Chang SL et al. demonstrated the effect of IL-17 on cardiac ventricular fibrosis via activating the MAPK pathway in rabbit ischemic heart failure models [50].

### 4.4. Inspiration and Reflection on Electroimmunological Mechanism

In conclusion, immune reactions after MI can directly or indirectly regulate ventricular electrical remodeling. The former mentioned electroimmunological regulatory mechanisms, mostly based on studies of normal cardiomyocytes; research on electroimmunology in the MI model was still limited. A 4-day course of IL-1β inhibition treatment with anakinra could improve conduction velocity, reduce APD dispersion, and enhance intracellular Ca^2+^ handling at day 5 post-MI, which was linked to increased expression of Cx43 and sarcoplasmic reticulum Ca^2+^-ATPase [51]. Another study found that continuous injection of the caspase-1 inhibitor VX765 upregulates Cx43 expression and improves cell-to-cell communication in rat hearts on the seventh day post-MI via suppressing the IL-1β/p38 MAPK pathway [52]. Researchers used a recombinant human tumor necrosis factor receptor: Fc fusion protein to neutralize TNF-α 24 h before the induction of MI, which finally reduced the incidence of VAs within 12 h after MI [53]. Most studies focus only on structural remodeling induced by MI, which indirectly reflects anti-inflammatory therapeutic benefits of anti-arrhythmia by promoting myocardial fibrosis. Due to the spatiotemporal heterogeneity of the inflammatory response following MI, anti-inflammatory therapy requires the selection of appropriate patient populations, optimal time windows, and specific molecular targets to reduce side effects. Sorting through the current clinical research findings on anti-inflammatory treatment for MI, it is found that patients with a high inflammatory burden who receive early anti-inflammatory treatment with IL-1α and IL-6 (3–24 h) can achieve a favorable prognosis. The occurrence period of arrhythmia is roughly the same as the staging of acute MI. So, corresponding to the three stages of arrhythmias induced by MI, anti-inflammatory methods through inhibiting the effects of cytokines like TNF-α, IL-6, and IL-1β, can reduce the occurrence of VAs in Stage I and Stage II by ameliorating the re-entry mechanism and abnormal automaticity, meanwhile the susceptibility to arrhythmias in Stage III is also reduced due to the improvement in fibrosis. Infection and tumor often occur concomitantly with immunosuppressive therapy. Is the current situation of anti-inflammatory treatment in the process of MI the same? Are the side effects, such as infection caused by immunosuppressive therapy, inevitable? According to the clinical research results on anti-inflammatory treatment for MI, pulmonary infections caused by the treatment can be mostly resolved by anti-infective treatment and drug withdrawal, with no reports of tumor-related side effects. Identifying new targets for evaluating the inflammatory burdens of patients with MI and pathogenic cytokines is crucial for the precise treatment of MI patients. In addition to the above targets, IL-17 and other cytokines have good interventional values, which need more clinical trials to validate their effectiveness. Besides, finding an effective and highly selective battleground for anti-inflammatory therapy is also a good solution. Cardiac ANS may be a potential choice.

## 5. Neuromodulation Mechanism of MI

The heart is precisely regulated beat by beat through the cardiac ANS. The ANS consists of the sympathetic and parasympathetic nervous systems. When the outflow of SNS increases, the heartbeat accelerates, and the cardiac output is also elevated. At the same time, the excitation of the parasympathetic nervous system inhibits cardiac function, antagonizing the effect of SNS. The imbalance between the sympathetic and parasympathetic nervous systems, in other words, sympathetic overactivation and parasympathetic withdrawal, is characteristic of an ANS induced by MI. Moreover, it is evident that both sympathetic tension inhibition and parasympathetic tension excitation can reduce the occurrence of VAs post-MI, as demonstrated in pre-clinical and clinical research [17]. ANS impacts the excitability and conduction of the heart by releasing neurotransmitters. At the cellular level, sympathetic activation modulates trigger activity by handling ion currents. The overflow of norepinephrine (NE) combined with the adrenergic receptor allows for more Ca^2+^ influx into the cytoplasm through the reactivation of L-type Ca^2+^ channels, the phosphorylation of phospholamban, and RyR [17]. Calcium overloading in the endoplasmic reticulum prolongs the duration of Phase 2 AP, predisposing the cardiomyocytes to a more susceptible alternant situation. Additionally, I_Na_ can also be activated by sympathetic activation, leading to increased AP amplitude and velocity and APD prolongation [17]. During sustained sympathetic activation, I_K, S_ was increased, leading to the shortening of APD and ERP [17]. Regarding tissue level, SNS overactivation is spatially heterogeneous [23]. Spatial heterogeneity makes the heart more conducive to forming re-entry circuits. Regrettably, clinical patients receiving adequate β-blockers still develop VAs or cardiac failure, which makes researchers believe that there must exist catecholamine-independent mechanisms modulating cardiomyocytes and highlights the importance of exploring the mechanism of VAs induced by sympathetic overactivation and the mechanism of sympathetic overactivation evoked by MI.

### 5.1. Myocardium in Response to Sympathetic Overactivation

Researchers found that chronic sympathetic overexcitation leads to a decrease in myocardial sensitivity to NE, mediated by the elevation of G-protein receptor kinase 2 (GRK2). GRK2 acts to inhibit β-AR activity in the form of a self-constrained negative feedback loop [54]. Additionally, sustained activation of cardiac sympathetic nerves leads to the downregulation of β-AR itself [55]. Those changes are adaptive to limit the sensitivity of cardiomyocytes in response to adrenergic stimulation. On the contrary, the infarcted area in chronic MI loses sympathetic innervation. Regional sympathetic hypoinnervation could increase the sensitivity of β-adrenergic receptors [56]. In addition, a recent study by Chen N. and colleagues demonstrated that cardiac sympathetic activation promotes copper overload in cardiomyocytes by inhibiting VPS35 expression [57].

### 5.2. Sympathetic Remodeling Induced by MI

Sympathetic remodeling after MI presents the following aspects: morphology, function, and innervation. In animal and human studies, both sympathetic ganglia, such as stellate ganglia, and parasympathetic ganglia, like nodose ganglia (NG) and the intrinsic cardiac nervous system, all remodel after MI. Neurons in SG enlarge and undergo cholinergic transdifferentiation [17].

#### 5.2.1. The Changes in Sympathetic Function

##### Sympathetic Transdifferentiation

Sympathetic transdifferentiation means sympathetic fibers release both NE and acetylcholine (ACh). Olivas et al. measured the peri-infarcted region and found that ACh levels increased significantly 10 and 14 days post-MI, then returned to sham levels 21 days after MI. While knocking out the ACh in SG neurons, ACh levels were similar to levels in the sham group. Moreover, the authors found that this transdifferentiation might originate from the stimulation of inflammatory cytokines, mediated by gp130, similar to the mechanism in nonischemic heart failure. To investigate whether this differentiation benefited the adaptation of the heart to ischemia, they further examined the APD and Ca^2+^ transient. They found that sympathetic transdifferentiation impaired the infarcted heart’s ability to adapt to higher heart rates [58]. Therewith, this team used optical mapping to detect the changes in cardiac electrophysiological properties after ACh knockout in noradrenergic neurons and found that sympathetic nerves co-releasing ACh could decrease the dispersion of AP, which may be antiarrhythmic [59] (shown in Figure 2).

##### Neurotransmitter

Another functional transformation is neurotransmitter release. David J. Paterson’s team found that an increased cAMP-PKA/cGMP-PKG ratio exacerbated cardiac sympathetic activity in hypertension, in which nitric oxide coupled to the nNOS adaptor protein could increase cGMP-PKG pathway flux; then, researchers overexpressed *nos1* in stellate neurons, which inhibited Ca^2+^ influx to the cytoplasm and restored Ca^2+^-dependent exocytosis in a model of spontaneously hypertensive [60]. cAMP-PKA/cGMP-PKG also plays an important role in regulating Ca^2+^ flux into the myocardium. Upregulating NO-CAPON in the myocardium remarkably decreased cGMP-PKG-related Ca^2+^ and K^+^ flux by gene transfer and shortened the QT interval [61]. Furthermore, phosphodiesterase enzymes (PDEs) are at the center of the balance of cyclic nucleotides (CNs); however, it is difficult to intervene with PDEs to regulate CNs because there is little chance to precisely target PDE isoforms for their specific distribution in cell compartments. Besides, they also emphasized that the production and release of adrenaline from presynaptic sympathetic nerve fibers were regulated with positive feedback by β2-adrenoceptor activation in prehypertensive rats [62]. Herring’s team found that neuropeptide Y (NPY) was another critical sympathetic co-transmitter that regulates cardiomyocytes. NPY elevates intracellular Ca^2+^ concentration by binding to the Y1 receptor on cardiomyocytes [63]. Kalla M showed that NPY could independently influence the arrhythmogenic properties of isolated rat hearts. Furthermore, adding the Y1 receptor inhibitor could increase the VF threshold in response to burst pacing on the stellate ganglion(SG) in the presence of a β-receptor blocker [64]. In addition, some other co-transmitters, like galanin [65], and ATP [66], were found in sympathetic nerve fibers. Whether there exist other molecules acting like neurotransmitters and whether those molecules play a different role in diverse CVDs needs more exploration (shown in Figure 2).

##### Sympathetic Overactivation

Sympathetic hyperactivation after MI has been widely confirmed in clinical and preclinical studies. By directly recording the cardiac sympathetic nerve discharge in Beagle dogs after MI, it was found that the frequency and amplitude of nerve discharge significantly increased 15 s before the occurrence of VF [67]. Studies have shown that the increase in cardiac sympathetic output following MI can persist for several months [68]. The excited sympathetic nerve can directly regulate the electrophysiological properties of the myocardial tissue by releasing neurotransmitters to activate the corresponding receptors in the myocardium. Overall, sympathetic nerve activation increases myocardial automaticity, enhances triggering activity, aggravates the re-entry mechanism, shortens the myocardial APD and ERP, increases QT interval dispersion, reduces the VF threshold, and increases susceptibility to arrhythmias. This evidence is the basis for sympathetic nerve regulation in the treatment of VAs after MI (shown in Figure 2).

#### 5.2.2. Sympathetic Innervation

At the heart tissue level, sympathetic remodeling shows as heterogeneous nerve sprouting in normal areas and border zones and denervation in scar region co-existing at the surface of the injured heart [69]; both were demonstrated to be arrhythmogenic. Physiologically, Trk tyrosine kinase receptors and p75NTR balance regulate cardiac sympathetic fiber innervation. NGF, combining Trk tyrosine kinase receptors, promotes the extent of fibers, while proNGF, BDNF, and proBDNF combine to p75NTR constrain the axon extent. NGF had been found to be overexpressed in the non-infarcted areas and SG after MI, leading to cardiac hyperinnervation [69]. Chondroitin sulfate proteoglycans (CSPGs) were thought to be the mediator resulting in sympathetic hypoinnervation; the presence of CSPGs was negatively related to sympathetic innervation [70]. CSPGs restrain sympathetic innervation by combining with the protein tyrosine phosphatase receptor σ (PTPσ). Researchers restored innervation across the scar region by removing or inhibiting the PTPσ, and the restored innervation significantly promoted cardiomyocyte calcium handling, decreased the dispersion of APD, and reduced the occurrence of VAs induced by MI [71]. Sema3a is identified as a potent neural chemorepellent and directional guidance molecule for nerve fibers. Cardiomyocytes can release Sema3a to negatively regulate cardiac sympathetic innervation. More importantly, the expressive kinetics of Sema3a correspond to the gradient distribution of cardiac sympathetic innervation from the epicardial to the endocardial layer [22]. Overexpression of Sema3a in the SG [72] reduced the susceptibility to VAs induced by MI. Recently, Boiteux C and colleagues proposed Sema3a as a biomarker for primary VF in patients with ST-elevation MI [73]. T-M Lee found that selectively blocking endothelin receptors could inhibit infarction-induced sympathetic innervation via a PI3K/GSK-3β-dependent pathway [74]. Then, they discovered that oxidant release promoted sympathetic hyperinnervation post-MI, and chronic usage of N-acetylcysteine attenuated cardiac sympathetic hyperinnervation through the replenishment of glutathione [75]. Buttgereit J found that C-type natriuretic peptide/natriuretic peptide receptor B signaling inhibited cardiac sympathetic neurotransmission and autonomic function [76]. Lähteenvuo J found that VEGF-β overexpression in adult hearts induced the upregulation of Nr4a2, ATF6, and MANF, resulting in nerve growth [77]. Collectively, those studies suggested multiple mechanisms, such as oxidative stress and receptor-ligand signaling pathways, regulating the inhomogeneity of sympathetic innervation.

### 5.3. Inspiration and Reflection on Neuromodulation Mechanism

The evidence overwhelmingly indicates that sympathetic nerve remodeling plays an undeniable role in promoting VAs following MI, making the suppression of excessive sympathetic activation a critical therapeutic approach. Current clinical interventions include β-blockers, cardiac sympathetic denervation (CSD), stellate ganglion block (SGB), and thoracic epidural anesthesia (TEA). However, these methods have notable limitations. Pharmacological treatments, due to their broad suppression of the SNS, lead to depression, weight gain, headaches, hyperglycemia, diarrhea, dizziness, claudication, and bradycardia [78]. CSD can cause complications such as pneumothorax, Horner syndrome, Harlequin syndrome, compensatory hyperhidrosis, nerve plexus injuries, neuropathic pain, and localized temperature changes. SGB is associated with risks like neck hematoma, transient hoarseness, systemic local anesthetic toxicity, and lacks a standardized marker for successful blockade. TEA cannot be combined with antiplatelet or anticoagulant therapies due to the risk of irreversible nerve damage from epidural hematoma [79,80].

These clinical application situations pose challenges for the use of SNS regulation in the treatment of arrhythmias after MI. On the one hand, we need to screen the population through reliable predictors. On the other hand, for the screened population, SNS regulation treatment with appropriate timing, intensity, duration, and scope is required to ensure the effectiveness and safety of the treatment. Clinically, wearable skin sympathetic nerve detection systems and imaging methods can be utilized to detect sympathetic nerve activity and innervation, thereby screening the intervention population and guiding the parameters of SNS regulation. It is worth noting that arrhythmias in the early stage of MI are mainly regulated by sympathetic activation, and SGB and TEA with short-term efficacy for high-risk populations may be a good choice. Arrhythmias in the chronic stage are mainly affected by the uneven distribution of sympathetic nerves. At this time, implementing CSD for high-risk populations may be a more effective treatment approach. However, some problems require large-scale randomized controlled trials to establish robust clinical evidence for precision therapy; in the case of SGB and TEA, which are suitable for short-acting anesthetics, and which are suitable for long-acting anesthetics? CSD selectively and irreversibly removes major cardiac sympathetic nerves, but clinical practice lacks clear criteria for selecting eligible patients. Whether the efficacy of SGB or TEA could serve as a predictor for CSD candidacy remains unclear. Furthermore, no head-to-head comparative studies have evaluated the efficacy of different sympathetic interventions for VAs.

The effectiveness of the aforementioned interventions confirms their critical role in post-infarction sympathetic activation, yet clinical research can only mitigate population-level side effects, not the inherent limitations of the treatments themselves. Therefore, advancing molecular studies of sympathetic activation during MI and identifying novel regulatory targets are essential to mitigate adverse effects. Anatomically, the SG innervates the heart, skin, and paws. Using neural tracing, Ajijola OA’s team demonstrated that cardiac-projecting neurons are primarily localized in the cranio-medial region of the SG, while paw-innervating neurons are distributed broadly. Single-cell RNA sequencing and functional analysis revealed that the heart is innervated by three subtypes of noradrenergic neurons, distinguished by NPY expression, with NPY identified as crucial for maximal cardiac sympathetic excitation [81]. Excitingly, sympathetic remodeling and activation are regulated by multiple factors, including inflammatory cytokines and satellite glial cells. Animal studies have shown that inhibiting these regulatory pathways reduces post-infarction arrhythmias, offering promising therapeutic avenues for targeted intervention.

## 6. The Crosstalk of Inflammation and the SNS

There is growing evidence that the immune and nervous systems interact with each other in many diseases [82,83,84]. In CVD, it has been reported that inflammation exacerbates sympathetic remodeling; conversely, sympathetic activation or innervation also influences inflammation. According to cardiac SNS’s anatomy, the interaction between the two systems includes the central SNS, the secondary sympathetic ganglion, and the heart as the targeted organ of sympathetic innervation. The underlying mechanism of this interaction is based on the neuroendocrine function of neurons, cytokines, and chemokines released from immune cells; at the same time, neurons and immune cells also express the corresponding receptors separately.

### 6.1. Central SNS

The paraventricular nucleus (PVN) is the main area of the central SNS and projects fibers, directly or indirectly, via the rostral ventrolateral medulla, to sympathetic premotor neurons at the intermediolateral cell column of the spinal cord [80]. Studies focusing on the PVN of the hypothalamus regulating heart diseases have shown that microglia play an essential role in the modulation of neurons’ growth and activity. Peng found that microglia-derived PDGFB promoted neuronal potassium currents to suppress basal sympathetic tonicity and limit hypertension [85]. Recently, they reported that specific high capillary density, thin vessel diameter, and complex vascular topology in the hypothalamic PVN enhanced this area’s susceptibility to penetrating ATP into the extravascular space, where it combined with the P2Y12 receptor on microglia, initiating microglial inflammation and ultimately strengthening sympathetic overflow [86]. At the same time, studies of microglia, as the central nervous system’s resident macrophages, have demonstrated that P2X7 receptor [87], macrophage-induced type C lectin [88], and TLR4 [89] expressed on microglia were activated post-MI, which increased pro-inflammatory cytokines expression and ROS production, leading to exacerbated sympathetic overactivation. Qi used RNA-seq, MeRIP-seq, and MeRIP-qPCR to find that METTLE3-mediated m6A methylation of TLR4 induced sympathetic hyperactivity by combining to activate NF-κB [90].

### 6.2. Second Level of SNS Hierarchy

SG is a vital junction of the sympathetic efferent pathway. SG from patients with cardiomyopathy and refractory VAs showed inflammatory cell infiltration and glial activation, suggesting inflammation was related to SG remodeling [91]. Satellite glial cells (SGCs) are a counterpart of astrocytes in the central nervous system and envelop the neurons in the peripheral nervous system. Despite SGCs being electrically non-excitable, SGCs and neurons can mutually regulate each other via gap junctions and release signal mediators [92]. SGCs have been demonstrated to secrete NGF to modulate the restoration and activity of neurons [93]. Enes J and his colleagues recently revealed that SGCs activate postganglionic sympathetic neurons and promote neuron survival from death [94]. However, how SGCs in SG regulate sympathetic neurons is less known. Whether SGCs also activate neurons by secret signal molecules in SG and what those are need to be further researched. Van Weperen screened SGCs in SG by glial-specific transcripts, S100b, and FABP7, and performed single-cell RNA sequencing of mouse SG. They revealed that SGCs could be sub-typed into six groups, in which two mature populations of SGCs upregulated cholesterol metabolism, and pathways related to cellular stress responses were upregulated in aging SGCs [95]. This reminds us that cellular metabolism and stress responses may participate in regulating SG. Macrophages are another important immune cell in SG. Zhang microinjected clodronate liposomes into bilateral SG to deplete macrophages, and results demonstrated that depleting macrophages in SG alleviated sympathetic overactivation by attenuating neuroinflammation and inhibiting N-type Ca^2+^ currents. Therefore, the incidence of VAs in chronic heart failure induced by surgical ligation of the coronary artery was decreased [96]. Pro-inflammatory cytokines like IL-6 [97] and IL-17A [98] were also demonstrated to aggravate cardiac sympathetic overactivation and VAs post-MI. But how do inflammation cytolines exacerbate cardiac sympathetic overactivation? Cyclin-dependent kinase 5 (CDK5), a proline-directed serine/threonine kinase, was reported to be increased by cytokines. Meanwhile, CDK5 could phosphorylate the N-type Ca^2+^ channels, then aggravate sympathetic overactivation [99]. Obviously, those studies suggest inflammation in the SNS contributes to cardiac sympathetic hyperactivation.

### 6.3. Third Level of the SNS

At the level of the targeted organ, sympathetic hyperactivation can regulate inflammation in the heart. Lyu J had proven in vivo and in vitro experiments that catecholamines secreted from sympathetic nerve terminals could alter cardiac sympathetic innervation via β1-AR on macrophages. As a result, VAs induced by MI were alleviated by macrophage depletion with clodronate and β1-adrenergic blockade with metoprolol [100]. Sepe JJ used two therapeutics to stimulate the sympathetic reinnervation of the reperfusion heart, leading to a shift from inflammatory to reparative condition, with fewer pro-inflammatory macrophages and more regulatory T cells, and alleviating VAs induced by isoproterenol 2 weeks after myocardial ischemia-reperfusion [101]. However, another study performed by Ziegler KA demonstrated that bilateral superior cervical ganglionectomy eliminated almost all sympathetic innervation of the anterior wall of the left ventricle and attenuated myocardial inflammation primarily at the border zone 2 weeks after MI [102]. Interestingly, both sympathetic denervation and reinnervation may be anti-inflammatory. The former studies promoted sympathetic reinnervation by intraperitoneal injection of chondroitin sulfate proteoglycans inhibitors from the third day after myocardial ischemic-reperfusion. In contrast, the latter studies performed a ganglionectomy immediately after myocardial ischemia; therefore, the two studies’ pathological models and intervention times are different. In addition, the seemingly contradictory result provides important information that sympathetic fiber innervation after myocardial ischemia may play different roles in cardiac inflammation. In the early ischemic phase, sympathetic innervation is pro-inflammatory, but once it passes through the early phase it is anti-inflammatory. Besides macrophages, other immune cells exist, including T lymphocytes, dendritic cells, NK cells, etc. Whether cardiac sympathetic nerves can modulate them after MI is less well understood. This big gap in regulating inflammation by sympathetic innervation post-MI warrants further exploration.

### 6.4. Other Regulatory Factors

Immune cells in immune organs are innervated by sympathetic nerves, like the thymus, bone marrow (BM), spleen, mucosa-associated lymphoid tissue, and tertiary lymphoid organs. Recently, researchers began to pay attention to the effect of myocardial injury by regulating sympathetic innervation in other immune organs after MI. Sympathetic nerves from the celiac ganglia abundantly innervate the spleen. The selective left celiac ganglion denervation of the spleen demonstrated an excellent ability for anti-hypertensive effects [103]. Besides, the parasympathomimetic function also plays a vital role in the spleen. Researchers recently found that CD4^+^ T cells in the spleen could produce ACh and then modulate macrophages [104]. Similarly, sympathetic nerves also innervate BM. Dutta P. found that tyrosine hydroxylase (TH), the rate-limiting enzyme for producing noradrenaline in sympathetic fibers, was elevated in the BM after MI. They further proved SNS activity could regulate the release of hematopoietic stem and progenitor cells (HSPCs) from BM, as blood HSPCs were reduced after MI in mice treated with β3 receptor antagonist [105]. It is worth noting that inflammation is our body’s defender, protecting us from various endogenous and exogenous stimuli. It was also reported to be related to other factors, like lifestyle. Recently, a paper published in Circulation suggested adding ‘psychological health’ to ‘life essential 8’, including diet, physical activity, nicotine exposure, body mass index, blood lipids, blood glucose, and blood pressure, to form ‘life crucial 9’ [106]. It highlights that lifestyle is closely related to cardiovascular health. Huynh P. reported that patients or mice suffering disrupted sleep would experience more severe myocardial damage. Because augmented sleep could recruit circulating monocytes to the thalamic lateral posterior nucleus. Those recruited monocytes produced TNF and then increased deep sleep by employing glutamatergic neurons. Disrupting sleep could increase cardiac sympathetic output and activate macrophages at the heart to secrete chemokines by β2-AR on macrophages, aggravating cardiac injury [107]. Similarly, Chen found that sleep fragmentation exacerbated myocardial ischemia-reperfusion injury by promoting sympathetic overactivity [57]. In addition, there are a number of studies that have reported on the relationship between psychological health and CVD. According to statistics, approximately 20% of patients with coronary artery disease have depression [108]. The risk of cardiovascular death and all-cause mortality in stable coronary artery disease patients with persistent moderate or greater psychological distress was 3.94 times and 2.85 times respectively, in comparison with patients without distress [109]. There is limited clinical evidence supporting anti-depression therapeutics’ benefits for reducing the cardiovascular-related mortality of patients with MI [110], but inflammation and altered hypothalamic–pituitary–adrenal axis function are widely accepted to be their common mechanism [108]. Obviously, the modulation of inflammation and SNS modulation are the underlying mechanisms of the ‘life essentials’ of CVD. Exploration of the ‘life essentials’ mechanisms provides new insights and helps us identify new therapeutic targets for MI.

### 6.5. Opportunities and Challenges in the Clinical Translation of Neuroimmunotherapy

The three-level SNS hierarchy serves as a battleground for neuroimmune regulation. Although there were some preclinical explorations of neuroimmune regulation in CVDs, which presented good effects for controlling the ANS imbalance, the clinical translation needs more research on the mechanisms.

## 7. Conclusions

Excitability, trigger activity, and the re-entry mechanism are the cornerstones of arrhythmia induced by MI. Treatments focus on coronary arteries and myocardium, like percutaneous coronary intervention, and AADs are still the primary therapies in clinical practice. However, some patients do not respond to these therapeutics. This review first summarizes the electrophysiological mechanism and characteristics of VAs at different phases post-MI for their unshakable basic status. Then, we also discussed neuromodulation, the electroimmunological mechanism of VAs, and their interaction (shown in Figure 3). All three levels of the SNS hierarchy provide us with aimed sites for therapies, and immune mechanisms at different levels of the SNS provide us with precise intervention objects. Furthermore, cells like microglia in the PVN, SGCs in the sympathetic ganglia, macrophages in the heart, and cytokines like IL-1, IL-6, and so on are the potential intervention targets. However, there is a need for more verification of effectiveness and safety. In addition, keeping the concept of the whole organism, exploring the interaction of other sympathetic innervated organs like the spleen and BM, and combining other co-factors like sleep, psychological health, and physical activity with MI can provide us with different perspectives to find new therapeutic targets.

## Figures and Tables

**Figure 2 biomedicines-13-01290-f002:**
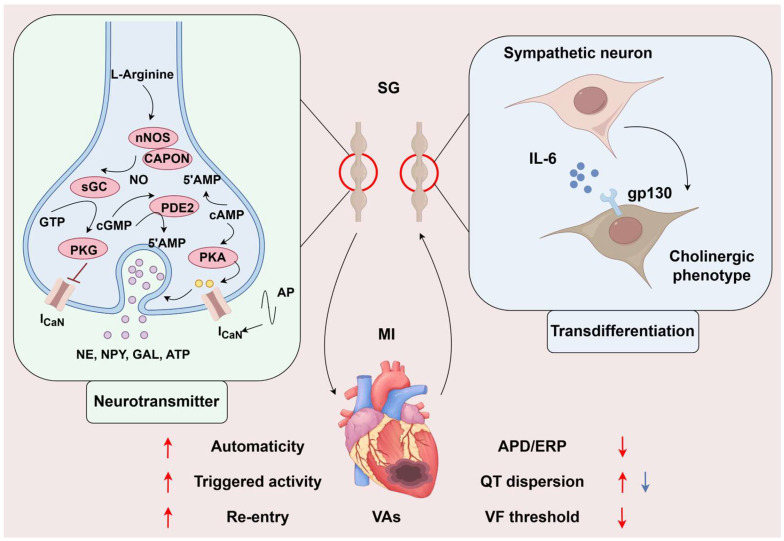
The changes in sympathetic function in myocardial infarction. The increased cAMP-PKA/cGMP-PKG ratio exacerbated cardiac sympathetic activity, leading to Ca^2+^ influx into neurons, promoting neurotransmitters to be released. Meanwhile, MI causes transient cholinergic transdifferentiation of cardiac sympathetic nerves via gp130. The former exacerbates ventricular electrophysiological instability and has a pro-arrhythmic effect, while the latter may stabilize ventricular electrophysiological stability and has an anti-arrhythmic effect. The role of neurotransmitters on ventricular electrophysiology is indicated by the red arrow, and the blue arrow indicates the role of transdifferentiation on ventricular electrophysiology. SG, stellate ganglion; MI, myocardial infarction; VAs, ventricular arrhythmias; AP, action potential; NE, norepinephrine; NPY, Neuropeptide Y; GAL, galanin; ATP, adenosine triphosphate; APD, action potential duration; ERP, effective refractory period; VF, ventricular fibrillation. (By Figdraw, https://www.figdraw.com/static/index.html#/, 11 March 2025).

**Figure 3 biomedicines-13-01290-f003:**
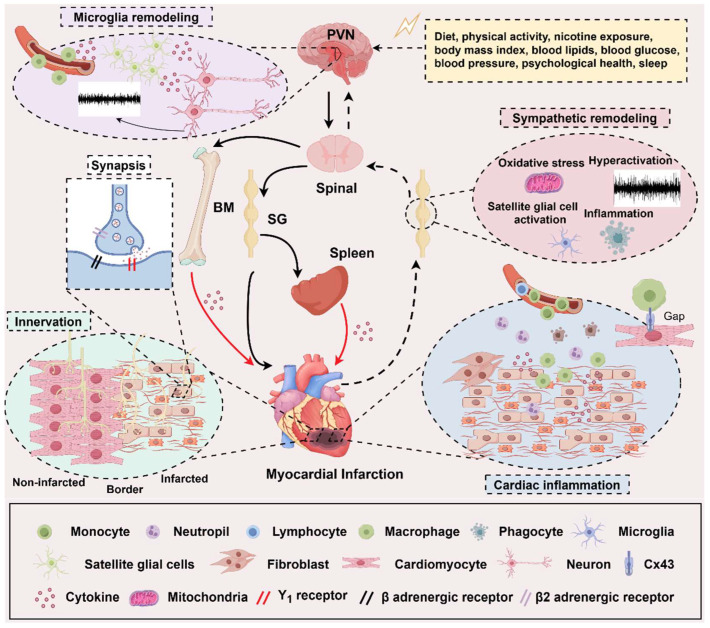
The neuroimmune mechanisms in myocardial Infarction. The regulation among the heart, the immune system, and the sympathetic nervous system (SNS) forms a complex network of cross-linking. On the one hand, the heart is regulated by the SNS from top to bottom and inflammation from the bone marrow (BM) and spleen. On the other hand, sympathetic nerves innervate both of them. From top to bottom, central sympathetic activity is influenced by various stimulating factors, and these stimuli can trigger microglial activation in the PVN area, thereby regulating central sympathetic neuron remodeling. The peripheral sympathetic ganglia are regulated by oxidative stress and the activation of satellite glial cells and inflammatory factors, resulting in an increase in sympathetic efferent that is relatively independent of central regulation. At the cardiac level, the myocardial tissue is affected by the neurotransmitters released by the efferent sympathetic nerve endings and the heterogeneous sympathetic nerve innervation, which exacerbates ventricular remodeling after MI. The immune cells derived from the resident myocardium, BM, and spleen can directly participate in the regulation of MI, on the one hand, and at the same time, they can also be regulated by the sympathetic nerve, thereby becoming a link in the middle of the SNS regulating MI. The solid black arrow represents a sympathetic efferent pathway; the black dotted arrow presents a visceral afferent pathway; the solid red arrow is an indirect effect of inflammation; PVN, paraventricular nucleus; BM, bone marrow; SG, sympathetic ganglia. (By Figdraw, https://www.figdraw.com/static/index.html#/, 11 March 2025).

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
