# Peer review of "Ventricular Arrhythmias and Myocardial Infarction: Electrophysiological and Neuroimmune Mechanisms"

_biomedicines, 2025, doi:10.3390/biomedicines13061290_

Round 1
Reviewer 1 Report
Comments and Suggestions for Authors
The manuscript provides a comprehensive review of the mechanisms underlying ventricular arrhythmias (VAs) following myocardial infarction (MI), focusing on electrophysiological, neuroimmune, and neuromodulation mechanisms. Below are some comments on the manuscript.
1. The manuscript effectively highlights the importance of understanding the interplay between electrophysiological mechanisms and neuroimmune pathways in VAs. However, the novelty of this review could be enhanced by including a more explicit comparison with prior similar reviews in the introduction. How does this paper uniquely contribute to advancing the field?
2. The review summarizes a vast body of literature, but the integration of studies could be improved. For instance, while cytokines such as IL-6 and TNF are mentioned multiple times, a clearer synthesis of their specific roles in different MI stages (acute, subacute, chronic) and their clinical implications would strengthen the discussion.
3. The manuscript discusses potential therapeutic targets but could benefit from more in-depth consideration of clinical translation. For example, while the authors propose immune modulation and ANS-targeted therapies, a discussion on challenges in clinical implementation (e.g., off-target effects, timing of interventions) would provide a balanced perspective.
4. The figures included in the manuscript (e.g., Figure 1, Figure 2) are well-designed but would be further improved with additional labeling or captions to better explain their relevance to the text. Consider expanding the legends to describe how the data presented in the figures ties into the key themes of the review.
Author Response
Response to reviewer’ comments
MS ID: Biomedicines-3431468
MS TITLE: Ventricular Arrhythmias and Myocardial Infarction: Electrophysiological and Neuroimmune Mechanisms
Thank you very much for taking the time to review this manuscript. Please find the detailed responses below.
Question-1 The manuscript effectively highlights the importance of understanding the interplay between electrophysiological mechanisms and neuroimmune pathways in VAs. However, the novelty of this review could be enhanced by including a more explicit comparison with prior similar reviews in the introduction. How does this paper uniquely contribute to advancing the field?
Response: Thanks for your constructive suggestions. Comparison with prior similar reviews [35642214, 33654273, 31197232, 36038114, 36201902, 37166736], our review not only concludes the electrical physiological mechanism of myocardial infarction but also elucidates the mechanisms of inflammation and autonomic neuromodulation of VAs post-MI. More importantly, we also discussed the interaction of immune cells at different levels of the sympathetic nervous system and incorporated the crosstalk between immune organs, lifestyle, and the nervous system. This will provide a multi-system interaction idea for exploring the mechanism of VAs induced by MI. Meanwhile, based on your opinions, we have added a comparison with similar reviews in the “Introduction” section and presented the novelty and significance of this review. Specifically, this is evident at the end of the “Introduction” section.
Question-2 The review summarizes a vast body of literature, but the integration of studies could be improved. For instance, while cytokines such as IL-6 and TNF are mentioned multiple times, a clearer synthesis of their specific roles in different MI stages (acute, subacute, chronic) and their clinical implications would strengthen the discussion.
Response: Thanks for your professional advice. Since both IL-6 and TNF are pro-inflammatory factors that mainly play a role in the early stage of myocardial infarction, at the same time, the existing studies on the relationship between inflammatory factors and arrhythmias are mostly based on the analogy of normal myocardial cells and tissues, and there are few direct studies on the relationship between inflammatory factors and arrhythmias after MI. Therefore, we did not introduce this in different periods after myocardial infarction in this manuscript. However, according to your suggestions, we have added a new part named “Inspiration and Reflection” to improve the integration of studies. This part illustrates some clinically relevant studies and deeply discusses the clinical transformation problem of anti-inflammatory treatment for ventricular arrhythmias after myocardial infarction. Specifically, this can be seen in the “4.4” section.
Question-3 The manuscript discusses potential therapeutic targets but could benefit from more in-depth consideration of clinical translation. For example, while the authors propose immune modulation and ANS-targeted therapies, a discussion on challenges in clinical implementation (e.g., off-target effects, timing of interventions) would provide a balanced perspective.
Response: Thank you for your pertinent suggestions. Firstly, this review focuses on introducing the basic research on the molecular mechanism of arrhythmia after myocardial infarction. We hope this review can offer new perspectives on the treatment of ventricular arrhythmia following myocardial infarction. Therefore, the clinically relevant studies are not the key points to be introduced. However, based on your opinions and after comprehensive consideration, we have added the content of "Inspiration and Reflection" in sections “4.4.” and “5.3.” respectively, which discussed the clinical transformation issues of immune and autonomic nervous modulation therapy in MI (including off-target effects, timing of interventions). Specifically, it can be seen in the corresponding part of the manuscript.
Question-4 The figures included in the manuscript (e.g., Figure 1, Figure 2) are well-designed but would be further improved with additional labeling or captions to better explain their relevance to the text. Consider expanding the legends to describe how the data presented in the figures ties into the key themes of the review.
Response: Thank you for your professional suggestions. According to your suggestions, we have revised Fig. 1 and Fig. 2. To enhance the connection between the figures and the text, we have rewritten the corresponding legends to describe the content of the figures, changed the position of figures in the text, and added bold notations (e.g., shown in Fig. 1) in the text to facilitate readers in locating and understanding the figure content. Specifically, it can be seen in the manuscript.
Reviewer 2 Report
Comments and Suggestions for Authors
The manuscript presents a review, which is focused on the electrophysiological, neuromodulation, and electroimmunological regulation mechanisms of ventricular arrhythmias induced by myocardial infarction. The review is interesting and worth to be published, since it presents a lot of data and refers considerable amount of information sources. However, I miss the description of the methodology applied for the design of the review – i.e. the criteria for selection of papers. Here below are my recommendations:
1) At the end of section ‘Introduction’, the authors should clearly state the question to which this review provides an answer.
2) The authors should add a new section ‘Methods’ just after the ‘Introduction’, where they should describe the applied methodology for paper selection – i.e. used scientific databases, pre-set period of interest, language of the papers, used keywords for search, inclusion and exclusion criteria, etc. Here are two possible sources (but not the only one) that could be followed in the process of writing a review:
- https://familymedicine.med.wayne.edu/mph/project/green_2006_narrative_literature_reviews.pdf
- https://www.theneuron.ai/write/reviews/narrative-review
3) It is noticeable that some parts are written in good English language, but others are not. In this respect I strongly recommend the authors to rewrite section ‘Introduction’.
Minor remark:
- The following sentence does not have a verb: “RMP, rest membrane potential; RyR, ryanodine receptors; NCX, Na+/Ca2+-exchanger; SERCA, sarcoplasmic reticulum Ca2+- ATPase; APD, action potential duration; MI, myocardial infarction.” If it is part from the caption of Fig. 1 it should be written in the caption, otherwise it should be corrected.
- The following sentence does not have a verb: “Hoping those mechanisms pro-20 vide new enlightenment for comprehending the regulatory mechanisms of VAs induced 21 by MI.”
Comments on the Quality of English LanguageIt is noticeable that some parts are written in good English language, but others are not.
Author Response
Response to reviewer’ comments
MS ID: Biomedicines-3431468
MS TITLE: Ventricular Arrhythmias and Myocardial Infarction: Electrophysiological and Neuroimmune Mechanisms
Thank you very much for taking the time to review this manuscript. Please find the detailed responses below.
Question-1 At the end of section ‘Introduction’, the authors should clearly state the question to which this review provides an answer.
Response: Thank you for your constructive opinions. We have modified this part according to your opinions. We have put forward the problems as follows: An in-depth exploration of the mechanisms of occurrence and maintenance of ventricular arrhythmias after MI, as well as the search for new therapeutic targets, is a breakthrough in the existing treatment model for arrhythmias. We have highlighted the significance of this review as follows: Unlike existing syntheses, this review innovatively integrates recent breakthroughs in cardiac neuroimmunology and neuromodulation studies, emphasizing the crosstalk between electrophysiology and neuroimmunology, which is expected to provide both timely and transformative insights for VA management paradigms. For brevity, the complete reasoning is outlined in the manuscript's “Introduction” section.
Question-2 The authors should add a new section ‘Methods’ just after the ‘Introduction’, where they should describe the applied methodology for paper selection – i.e. used scientific databases, pre-set period of interest, language of the papers, used keywords for search, inclusion and exclusion criteria, etc. Here are two possible sources (but not the only one) that could be followed in the process of writing a review:
- https://familymedicine.med.wayne.edu/mph/project/green_2006_narrative_literature_reviews.pdf
-https://www.theneuron.ai/write/reviews/narrative-review
Response: Thank you for your rigorous suggestions. According to your suggestions, we have added the methodological content after the “Introduction” part of the manuscript. The modified content is as follows:
Methods: The authors conducted a systematic search across three databases: PubMed, Web of Science, and Google Scholar. The searching span from January 1, 2016, to January 1, 2025. The searching terms included “Myocardial infarction” (â‘ ), “myocardial ischemia” (â‘ ), “ventricular arrhythmias” (â‘¡), “ventricular tachycardia” (â‘¡), “ventricular fibrillation” (â‘¡), “inflammation” (â‘¢), “immune” (â‘¢), “autonomic nervous system” (â‘£), “sympathetic nervous system” (â‘£),s “neuroimmune” (⑤), and “neuroinflammation” (⑤). The search strategy is “â‘ and â‘¡ and â‘¢” or “â‘ and â‘¡ and â‘£” or “â‘ and â‘¡ and ⑤”. In addition, important references from the search papers have also been included. Articles related to the field of search terms were included in this review, except for articles not available in English.
Question-3 It is noticeable that some parts are written in good English language, but others are not. In this respect I strongly recommend the authors to rewrite section ‘Introduction’.
Minor remark:
- The following sentence does not have a verb: “RMP, rest membrane potential; RyR, ryanodine receptors; NCX, Na+/Ca2+-exchanger; SERCA, sarcoplasmic reticulum Ca2+- ATPase; APD, action potential duration; MI, myocardial infarction.” If it is part from the caption of Fig. 1 it should be written in the caption, otherwise it should be corrected.
- The following sentence does not have a verb: “Hoping those mechanisms pro-20 vide new enlightenment for comprehending the regulatory mechanisms of VAs induced 21 by MI.”
Response: Thank you for your careful review and pertinent suggestions. We have revised the grammatical content of the entire text in accordance with your suggestions and rewritten the “Introduction” section. The details can be seen in the original text. “RMP, rest membrane potential; RyR, ryanodine receptors; NCX, Na+/Ca2+-exchanger; SERCA, sarcoplasmic reticulum Ca2+-ATPase; APD, action potential duration; MI, myocardial infarction.” is part of the caption of Fig. 1, so we rewrote it in the caption. Due to the modification of the “Introduction” part, the sentence “Hoping those mechanisms pro-20 vide new enlightenment for comprehending the regulatory mechanisms of VAs induced 21 by MI” has been deleted.
Reviewer 3 Report
Comments and Suggestions for Authors
The topics addressed in the review are of great interest, and it is evident that the authors have a profound knowledge of the subject. However, the article is somewhat challenging to read, and the work could be better organized to facilitate the reader, especially in terms of visual presentation.
- In the introduction, the statement "Collectively, those emphasize that VAs are the prevention and treatment key for MI" represents an incorrect assertion. The studies cited, as presented, indicate a correlation between ventricular arrhythmias and post-infarction mortality, but not a cause-effect relationship. It is necessary to better explain the scientific evidence underlying this statement or to eliminate this deduction altogether.
- Figure 1 is indecipherable. There is no caption to clarify its content, and the relationship between the different panels is unclear. Additionally, the graphical quality is poor and should be improved globally.
- It is recommended to create a summary figure or diagram for paragraph 2.2.1.1 (covering both action potential duration and triggers) and a separate figure for paragraph 4.2.1.
- There are too many citations for a narrative review. A selection should be made to reduce them by at least one-third
Author Response
A point-by-point response to the reviewer can be found in the attachment named "Response to reviewer 3-Biomedicines-3431468"

Round 2
Reviewer 1 Report
Comments and Suggestions for Authors
The authors have addressed all my previous concerns appropriately. I find the revised manuscript acceptable, and I have no additional suggestions at this time.
Author Response
Response to reviewer’s comments
MS ID: Biomedicines-3431468
MS TITLE: Ventricular Arrhythmias and Myocardial Infarction: Electrophysiological and Neuroimmune Mechanisms
Thank you very much for taking the time to review this manuscript. Please find the detailed responses below.
Question: The authors have addressed all my previous concerns appropriately. I find the revised manuscript acceptable, and I have no additional suggestions at this time.
Response: It is precisely because of your constructive comments on this review that the innovation of the article, the relevance of each part, and the significance of the review have been enhanced. Thank you for agreeing to accept this review.
Reviewer 2 Report
Comments and Suggestions for Authors
The manuscript has been adequately revised according to the recommendations in my first report. I have no other recommendations and in my opinion the manuscript is suitable for publication in its present form.
Author Response
Response to reviewer’s comments
MS ID: Biomedicines-3431468
MS TITLE: Ventricular Arrhythmias and Myocardial Infarction: Electrophysiological and Neuroimmune Mechanisms
Thank you very much for taking the time to review this manuscript. Please find the detailed responses below.
Question: The manuscript has been adequately revised according to the recommendations in my first report. I have no other recommendations and in my opinion the manuscript is suitable for publication in its present form.
Response: Thank you for your valuable suggestions on this review. As a result, the significance, integrity, and graphic-text relevance of this review have been enhanced. Thank you for agreeing to the acceptance of this review.